# Are deep ResNets provably
# better than linear predictors?

**Chulhee Yun**
MIT
Cambridge, MA 02139
chulheey@mit.edu

**Suvrit Sra**
MIT
Cambridge, MA 02139
suvrit@mit.edu

**Ali Jadbabaie**
MIT
Cambridge, MA 02139
jadbabai@mit.edu

## Abstract

Recent results in the literature indicate that a residual network (ResNet) composed of a single residual block outperforms linear predictors, in the sense that all local minima in its optimization landscape are at least as good as the best linear predictor. However, these results are limited to a single residual block (i.e., shallow ResNets), instead of the deep ResNets composed of multiple residual blocks. We take a step towards extending this result to deep ResNets. We start by two motivating examples. First, we show that there exist datasets for which all local minima of a fully-connected ReLU network are *no better* than the best linear predictor, whereas a ResNet has *strictly better* local minima. Second, we show that even at the global minimum, the representation obtained from the residual block outputs of a 2-block ResNet do not necessarily improve monotonically over subsequent blocks, which highlights a fundamental difficulty in analyzing deep ResNets. Our main theorem on deep ResNets shows under simple geometric conditions that, any critical point in the optimization landscape is either (i) *at least as good as* the best linear predictor; or (ii) the Hessian at this critical point has a *strictly* negative eigenvalue. Notably, our theorem shows that a chain of multiple skip-connections can improve the optimization landscape, whereas existing results study direct skip-connections to the last hidden layer or output layer. Finally, we complement our results by showing benign properties of the "near-identity regions" of deep ResNets, showing depth-independent upper bounds for the risk attained at critical points as well as the Rademacher complexity.

## 1 Introduction

Empirical success of deep neural network models has sparked a huge interest in the theory of deep learning, but a concrete theoretical understanding of deep learning still remains elusive. From the optimization point of view, the biggest mystery is why gradient-based methods find close-to-global solutions despite nonconvexity of the empirical risk.

There have been several attempts to explain this phenomenon by studying the loss surface of the risk. The idea is to find benign properties of the empirical or population risk that make optimization easier. So far, the theoretical investigation as been mostly focused on vanilla fully-connected neural networks [1, 8, 10, 11, 18, 20, 22–29, 31]. For example, Kawaguchi [8] proved that "local minima are global minima" property holds for squared error empirical risk of linear neural networks (i.e., no nonlinear activation function at hidden nodes). Other results on deep linear neural networks [10, 27, 29, 31] have extended [8]. However, it was later theoretically and empirically shown that "local minima are global minima" property no longer holds in nonlinear neural networks [20, 29] for general datasets and activations.

Moving beyond fully-connected networks, there is an increasing body of analysis dedicated to studying residual networks (ResNets). A ResNet [6, 7] is a special type of neural network that gained widespread popularity in practice. While fully-connected neural networks or convolutional neural networks can be viewed as a composition of nonlinear layers $x \mapsto \Phi(x)$, a ResNet consists of a series of **residual blocks** of the form $x \mapsto g(x + \Phi(x))$, where $\Phi(x)$ is some feedforward neural network and $g(\cdot)$ is usually taken to be identity [7]. Given these *identity skip-connections*, the output of a residual block is a feedforward network $\Phi(x)$ plus the input $x$ itself, which is different from fully-connected neural networks. The motivation for this architecture is to let the network learn only the *residual* of the input.

ResNets are very popular in practice, and it has been argued that they have benign loss landscapes that make optimization easier [12]. Recently, Shamir [21] showed that ResNets composed of a single residual block have "good" local minima, in the sense that any local minimum in the loss surface attains a risk value at least as good as the one attained by the best linear predictor. A subsequent result [9] extended this result to non-scalar outputs, with weaker assumptions on the loss function. However, these existing results are limited to a *single* residual block, instead of deep ResNets formed by composing multiple residual blocks. In light of these results, a natural question arises: *can these single-block results be extended to multi-block ResNets?*

There are also another line of works that consider network architectures with "skip-connections." Liang et al. [13, 14] consider networks of the form $x \mapsto f_S(x) + f_D(x)$ where $f_S(x)$ is a "shortcut" network with one or a few hidden nodes, and they show that under some conditions this shortcut network eliminates spurious local minima. Nguyen et al. [19] consider skip-connections from hidden nodes to the output layer, and show that if the number of skip-connections to output layer is greater than or equal to the dataset size, the loss landscape has no spurious local valleys. However, skip-connections in these results are all connections *directly to output*, so it remains unclear whether *a chain of multiple skip-connections can improve the loss landscape.*

There is also another line of theoretical results studying what happens in the **near-identity regions** of ResNets, i.e., when the residual part $\Phi$ is "small" for all layers. Hardt and Ma [5] proved that for linear ResNets $x \mapsto (I + A_L) \cdots (I + A_1)x$, any critical point in the region $\{\|A_l\| < 1 \text{ for all } l\}$ is a global minimum. The authors also proved that any matrix $R$ with positive determinant can be decomposed into products of $I + A_l$, where $\|A_l\| = O(1/L)$. Bartlett et al. [3] extended this result to nonlinear function space, and showed similar expressive power and optimization properties of near-identity regions; however, their results are on function spaces, so they don't imply that the same properties hold for parameter spaces. In addition, an empirical work by Zhang et al. [30] showed that initializing ResNets in near-identity regions also leads to good empirical performance. For the residual part $\Phi$ of each block, they initialize the last layer of $\Phi$ at zero, and scale the initialization of the other layers by a factor inversely proportional to depth $L$. This means that each $\Phi$ at initialization is zero, hence the network starts in the near-identity region. Their experiments demonstrate that ResNets can be stably trained without batch normalization, and trained networks match the *generalization* performance of the state-of-the-art models. These results thus suggest that understanding optimization and generalization of ResNets in near-identity regions is a meaningful and important question.

## 1.1 Summary of contributions

This paper takes a step towards answering the questions above. In Section 3, we start with two motivating examples showing the advantage of ResNets and the difficulty of deep ResNet analysis:

▶ The first example shows that there exists a family of datasets on which the squared error loss attained by a fully-connected neural network is at best the linear least squares model, whereas a ResNet attains a strictly better loss than the linear model. This highlights that the guarantee on the risk value of local minima is indeed special to residual networks.

▶ In the single-block case [21], we have seen that the "representation" obtained at the residual block output $x + \Phi(x)$ has an improved linear fit compared to the raw input $x$. Then, in multi-block ResNets, do the representations at residual block outputs improve *monotonically* over subsequent blocks as we proceed to the output layer? The second example shows that it is not necessarily the case; we give an example where the linear fit with representations by the output of residual blocks does not monotonically improve over blocks. This highlights the difficulty of ResNet analysis, and shows that [21] cannot be directly extended to multi-block ResNets.

Using new techniques, Section 4 extends the results in [21] to deeper ResNets, under some simple geometric conditions on the parameters.

▶ We consider a deep ResNet model that subsumes [21] as a special case, under the same assumptions on the loss function. We prove that if two geometric conditions called "representation coverage" and "parameter coverage" are satisfied, then a critical point of the loss surface satisfies at least one of the following: 1) the risk value is no greater than the best linear predictor, 2) the Hessian at the critical point has a strictly negative eigenvalue. We also provide an architectural sufficient condition for the parameter coverage condition to hold.

Finally, Section 5 shows benign properties of deep ResNets in the near-identity regions, in both optimization and generalization aspects. Specifically,

▶ In the absence of the geometric conditions above, we prove an upper bound on the risk values at critical points. The upper bound shows that if each residual block is close to identity, then the risk values at its critical points are not too far from the risk value of the best linear model. Crucially, we establish that the distortion over the linear model is independent of network size, as long as each blocks are near-identity.

▶ We provide an upper bound on the Rademacher complexity of deep ResNets. Again, we observe that in the near-identity region, the upper bound is independent of network size, which is difficult to achieve for fully-connected networks [4].

## 2  Preliminaries

In this section, we briefly introduce the ResNet architecture and summarize our notation.

Given positive integers $a$ and $b$, where $a < b$, $[a]$ denotes the set $\{1, 2, \ldots, a\}$ and $[a : b]$ denote $\{a, a+1, \ldots, b-1, b\}$. Given a vector $x$, $\|x\|$ denotes its Euclidean norm. For a matrix $M$, by $\|M\|$ and $\|M\|_{\mathrm{F}}$ we mean its spectral norm and Frobenius norm, respectively. Let $\lambda_{\min}(M)$ be the minimum eigenvalue of a symmetric matrix $M$. Let $\mathrm{col}(M)$ be the column space of a matrix $M$.

Let $x \in \mathbb{R}^{d_x}$ be the input vector. We consider an $L$-block ResNet $f_{\boldsymbol{\theta}}(\cdot)$ with a linear output layer:
$$h_0(x) = x,$$
$$h_l(x) = h_{l-1}(x) + \Phi_{\boldsymbol{\theta}}^l(h_{l-1}(x)), \quad l = 1, \ldots, L,$$
$$f_{\boldsymbol{\theta}}(x) = \boldsymbol{w}^T h_L(x).$$
We use bold-cased symbols to denote network parameter vectors/matrices, and $\boldsymbol{\theta}$ to denote the collection of all parameters. As mentioned above, the output of $l$-th residual block is the input $h_{l-1}(x)$ plus the output of the "residual part" $\Phi_{\boldsymbol{\theta}}^l(h_{l-1}(x))$, which is some feedforward neural network. The specific structure of $\Phi_{\boldsymbol{\theta}}^l : \mathbb{R}^{d_x} \mapsto \mathbb{R}^{d_x}$ considered will vary depending on the theorems. After $L$ such residual blocks, there is a linear fully-connected layer parametrized by $\boldsymbol{w} \in \mathbb{R}^{d_x}$, and the output of the ResNet is scalar-valued.

Using ResNets, we are interested in training the network under some distribution $\mathcal{P}$ of the input and label pairs $(x, y) \sim \mathcal{P}$, with the goal of minimizing the loss $\ell(f_{\boldsymbol{\theta}}(x); y)$. More concretely, the risk function $\mathfrak{R}(\boldsymbol{\theta})$ we want to minimize is
$$\mathfrak{R}(\boldsymbol{\theta}) := \mathbb{E}_{(x,y)\sim\mathcal{P}} \left[ \ell(f_{\boldsymbol{\theta}}(x); y) \right],$$
where $\ell(p; y) : \mathbb{R} \mapsto \mathbb{R}$ is the loss function parametrized by $y$. If $\mathcal{P}$ is an empirical distribution by a given set of training examples, this reduces to an empirical risk minimization problem. Let $\ell'(\cdot; y)$ and $\ell''(\cdot; y)$ be first and second derivatives of $\ell$, whenever they exist.

We will state our results by comparing against the risk achieved by linear predictors. Thus, let $\mathfrak{R}_{\mathrm{lin}}$ be the risk value achieved by the best linear predictor:
$$\mathfrak{R}_{\mathrm{lin}} := \inf_{t \in \mathbb{R}^{d_x}} \mathbb{E}_{(x,y)\sim\mathcal{P}} \left[ \ell(t^T x; y) \right].$$

## 3  Motivating examples

Before presenting the main theoretical results, we present two motivating examples. The first one shows the advantage of ResNets over fully-connected networks, and the next one highlights that deep ResNets are difficult to analyze and techniques from previous works cannot be directly applied.

Table 1: Lower bounds on $\mathfrak{R}_1(\boldsymbol{\theta}_1^*)$, if $w_1^* > 0$

| $-b_1^*/w_1^*$ in: | Error by constant part | Error by linear part | Lower bound |
|---|---|---|---|
| $(-\infty, 0)$ | 0 | $8\rho^2/15$ | $8\rho^2/15$ |
| $[0, 1)$ | 0 | $8\rho^2/15$ | $8\rho^2/15$ |
| $[1, 2)$ | $1/12$ | $7\rho^2/15$ | $7\rho^2/15 + 1/12$ |
| $[2, 3)$ | $4\rho^2/9 + 2\rho/3 + 1/3$ | $\rho^2/9$ | $5\rho^2/9 + 2\rho/3 + 1/3$ |
| $[3, 4)$ | $\rho^2/2 + \rho/3 + 5/6$ | 0 | $\rho^2/2 + \rho/3 + 5/6$ |
| $[4, 5)$ | $4\rho^2/5 + 4\rho/3 + 5/3$ | 0 | $4\rho^2/5 + 4\rho/3 + 5/3$ |
| $[5, \infty)$ | $\rho^2 + 7\rho/3 + 35/12$ | 0 | $\rho^2 + 7\rho/3 + 35/12$ |

### 3.1 All local minima of fully-connected networks can be worse than a linear predictor

Although it is known that local minima of 1-block ResNets are at least as good as linear predictors, can this property hold also for fully-connected networks? Can a local minimum of a fully-connected network be strictly worse than a linear predictor? In fact, we present a simple example where *all* local minima of a fully-connected network are at best as good as linear models, while a residual network has *strictly* better local minima.

Consider the following dataset with six data points, where $\rho > 0$ is a fixed constant:

$$X = [0 \quad 1 \quad 2 \quad 3 \quad 4 \quad 5], \quad Y = [-\rho \quad 1 - \rho \quad 2 + \rho \quad 3 - \rho \quad 4 + \rho \quad 5 + \rho].$$

Let $x_i$ and $y_i$ be the $i$-th entry of $X$ and $Y$, respectively. We consider two different neural networks: $f_1(x; \boldsymbol{\theta}_1)$ is a fully-connected network parametrized by $\boldsymbol{\theta}_1 = (w_1, w_2, b_1, b_2)$, and $f_2(x; \boldsymbol{\theta}_2)$ is a ResNet parametrized by $\boldsymbol{\theta}_2 = (w, v, u, b, c)$, defined as

$$f_1(x; \boldsymbol{\theta}_1) = w_2\sigma(w_1 x + b_1) + b_2, \quad f_2(x; \boldsymbol{\theta}_2) = w(x + v\sigma(ux + b)) + c,$$

where $\sigma(t) = \max\{t, 0\}$ is ReLU activation. In this example, all parameters are scalars.

With these networks, our goal is to fit the dataset under squared error loss. The empirical risk functions we want to minimize are given by

$$\mathfrak{R}_1(\boldsymbol{\theta}_1) := \frac{1}{6}\sum_{i=1}^{6}(w_2\sigma(w_1 x_i + b_1) + b_2 - y_i)^2, \ \mathfrak{R}_2(\boldsymbol{\theta}_2) := \frac{1}{6}\sum_{i=1}^{6}(w(x_i + v\sigma(ux_i + b)) + c - y_i)^2,$$

respectively. It is easy to check that the best empirical risk achieved by linear models $x \mapsto wx + b$ is $\mathfrak{R}_{\text{lin}} = 8\rho^2/15$. It follows from [21] that all local minima of $\mathfrak{R}_2(\cdot)$ have risk values at most $\mathfrak{R}_{\text{lin}}$. For this particular example, we show that the *opposite* holds for the fully-connected network, whereas for the ResNet there exists a local minimum *strictly* better than $\mathfrak{R}_{\text{lin}}$.

**Proposition 1.** *Consider the dataset $X$ and $Y$ as above. If $\rho \leq \sqrt{5/4}$, then any local minimum $\boldsymbol{\theta}_1^*$ of $\mathfrak{R}_1(\cdot)$ satisfies $\mathfrak{R}_1(\boldsymbol{\theta}_1^*) \geq \mathfrak{R}_{\text{lin}}$, whereas there exists a local minimum $\boldsymbol{\theta}_2^*$ of $\mathfrak{R}_2(\cdot)$ such that $\mathfrak{R}_2(\boldsymbol{\theta}_2^*) < \mathfrak{R}_{\text{lin}}$.*

**Proof** The function $f_1(x; \boldsymbol{\theta}_1)$ is piece-wise continuous, and consists of two pieces (unless $w_1 = 0$ or $w_2 = 0$). If $w_1 > 0$, the function is linear for $x \geq -b_1/w_1$ and constant for $x \leq -b_1/w_1$. For any local minimum $\boldsymbol{\theta}_1^*$, the empirical risk $\mathfrak{R}_1(\boldsymbol{\theta}_1^*)$ is bounded from below by the risk achieved by fitting the linear piece and constant piece separately, *without the restriction of continuity*. This is because we are removing the constraint that the function $f_1(\cdot)$ has to be continuous.

For example, if $w_1^* > 0$ and $-b_1^*/w_1^* = 1.5$, then its empirical risk $\mathfrak{R}_1(\boldsymbol{\theta}_1^*)$ is at least the error attained by the best constant fit of $(x_1, y_1), (x_2, y_2)$, and the best linear fit of $(x_3, y_3), \ldots, (x_6, y_6)$. For all possible values of $-b_1^*/w_1^*$, we summarize in Table 1 the lower bounds on $\mathfrak{R}_1(\boldsymbol{\theta}_1^*)$. It is easy to check that if $\rho \leq \sqrt{5/4}$, all the lower bounds are no less than $8\rho^2/15$. The case where $w_1^* < 0$ can be proved similarly, and the case $w_1^* = 0$ is trivially worse than $8\rho^2/15$ because $f_1(x; \boldsymbol{\theta}_1^*)$ is a constant function.

For the ResNet part, it suffices to show that there is a point $\boldsymbol{\theta}_2$ such that $\mathfrak{R}_2(\boldsymbol{\theta}_2) < 8\rho^2/15$, because then its global minimum will be strictly smaller than $8\rho^2/15$. Choose $v = 0.5\rho$,

$u = 1$, and $b = -3$. Given input $X$, the output of the residual block $x \mapsto x + v\sigma(ux + b)$ is $[0 \quad 1 \quad 2 \quad 3 \quad 4 + 0.5\rho \quad 5 + \rho] =: H$. Using this, we choose $w$ and $c$ that linearly fit $H$ and $Y$. Using the optimal $w$ and $c$, a straightforward calculation gives $\mathfrak{R}_2(\boldsymbol{\theta}_2) = \frac{\rho^2(12\rho^2 + 82\rho + 215)}{21\rho^2 + 156\rho + 420}$, and it is strictly smaller than $8\rho^2/15$ on $\rho \in (0, \sqrt{5/4}]$. $\qquad\square$

## 3.2 Representations by residual block outputs do not improve monotonically

Consider a 1-block ResNet. Given a dataset $X$ and $Y$, the residual block transforms $X$ into $H$, where $H$ is the collection of outputs of the residual block. Let $\mathrm{err}(X, Y)$ be the minimum mean squared error from fitting $X$ and $Y$ with a linear least squares model. The result that a local minimum of a 1-block ResNet is better than a linear predictor can be stated in other words: the output of the residual block produces a "better representation" of the data, so that $\mathrm{err}(H, Y) \le \mathrm{err}(X, Y)$.

For a local minimum of a $L$-layer ResNet, our goal is to prove that $\mathrm{err}(H_L, Y) \le \mathrm{err}(X, Y)$, where $H_l$, $l \in [L]$ is the collection of output of $l$-th residual block. Seeing the improvement of representation in 1-block case, it is tempting to conjecture that each residual block *monotonically improves* the representation, i.e., $\mathrm{err}(H_L, Y) \le \mathrm{err}(H_{L-1}, Y) \le \cdots \le \mathrm{err}(H_1, Y) \le \mathrm{err}(X, Y)$. Our next example shows that this monotonicity does not necessarily hold.

Consider a dataset $X = [1 \quad 2.5 \quad 3]$ and $Y = [1 \quad 3 \quad 2]$, and a 2-block ResNet

$$h_1(x) = x + v_1\sigma(u_1 x + b_1), \quad h_2(x) = h_1(x) + v_2\sigma(u_2 h_1(x) + b_2), \quad f(x) = wh_2(x) + c,$$

where $\sigma$ denotes ReLU activation. We choose

$$v_1 = 1,\ u_1 = 1,\ b_1 = -2,\ v_2 = -4,\ u_2 = 1,\ b_2 = -3.5,\ w = 1,\ c = 0.$$

With these parameter values, we have $H_1 = [1 \quad 3 \quad 4]$ and $H_2 = [1 \quad 3 \quad 2]$. It is evident that the network output perfectly fits the dataset, and $\mathrm{err}(H_2, Y) = 0$. Indeed, the chosen set of parameters is a global minimum of the squared loss empirical risk. Also, by a straightforward calculation we get $\mathrm{err}(X, Y) = 0.3205$ and $\mathrm{err}(H_1, Y) = 0.3810$, so $\mathrm{err}(H_1, Y) > \mathrm{err}(X, Y)$. This shows that the conjecture $\mathrm{err}(H_2, Y) \le \mathrm{err}(H_1, Y) \le \mathrm{err}(X, Y)$ is not true, and it also implies that an induction-type approach showing $\mathrm{err}(H_2, Y) \le \mathrm{err}(H_1, Y)$ and then $\mathrm{err}(H_1, Y) \le \mathrm{err}(X, Y)$ will never be able to prove $\mathrm{err}(H_2, Y) \le \mathrm{err}(X, Y)$.

In fact, application of the proof techniques in [21] only shows that $\mathrm{err}(H_2, Y) \le \mathrm{err}(H_1, Y)$, so a comparison of $\mathrm{err}(H_2, Y)$ and $\mathrm{err}(X, Y)$ does not follow. Further, our example shows that even $\mathrm{err}(H_1, Y) > \mathrm{err}(X, Y)$ is possible, showing that theoretically proving $\mathrm{err}(H_2, Y) \le \mathrm{err}(X, Y)$ is challenging even for $L = 2$. In the next section, we present results using new techniques to overcome this difficulty and prove $\mathrm{err}(H_L, Y) \le \mathrm{err}(X, Y)$ under some geometric conditions.

# 4  Local minima of deep ResNets are better than linear predictors

Given the motivating examples, we now present our first main result, which shows that under certain geometric conditions, each critical point of ResNets has benign properties: either (i) it is as good as the best linear predictor; or (ii) it is a *strict* saddle point.

## 4.1  Problem setup

We consider an $L$-block ResNet whose residual parts $\Phi_{\boldsymbol{\theta}}^l(\cdot)$ are defined as follows:

$$\Phi_{\boldsymbol{\theta}}^1(t) = \boldsymbol{V}_1 \phi_{\boldsymbol{z}}^1(t), \text{ and } \Phi_{\boldsymbol{\theta}}^l(t) = \boldsymbol{V}_l \phi_{\boldsymbol{z}}^l(\boldsymbol{U}_l t), \quad l = 2, \ldots, L.$$

We collect all parameters into $\boldsymbol{\theta} := (\boldsymbol{w}, \boldsymbol{V}_1, \boldsymbol{V}_2, \boldsymbol{U}_2, \ldots, \boldsymbol{V}_L, \boldsymbol{U}_L, \boldsymbol{z})$. The functions $\phi_{\boldsymbol{z}}^l : \mathbb{R}^{m_l} \to \mathbb{R}^{n_l}$ denote any arbitrary function parametrized by $\boldsymbol{z}$ that are differentiable almost everywhere. They could be fully-connected ReLU networks, convolutional neural networks, or any combination of such feed-forward architectures. We even allow different $\phi_{\boldsymbol{z}}^l$'s to share parameters in $\boldsymbol{z}$. Note that $m_1 = d_x$ by the definition of the architecture. The matrices $\boldsymbol{U}_l \in \mathbb{R}^{m_l \times d_x}$ and $\boldsymbol{V}_l \in \mathbb{R}^{d_x \times n_l}$ form linear fully-connected layers. Note that if $L = 1$, the network boils down to $x \mapsto \boldsymbol{w}^T(x + \boldsymbol{V}_1 \phi_{\boldsymbol{z}}^1(x))$, which is *exactly* the architecture considered by Shamir [21]; we are considering a deeper extension of the previous paper.

For this section, we make the following mild assumption on the loss function:

**Assumption 4.1.** *The loss function $\ell(p; y)$ is a convex and twice differentiable function of p.*

This assumption is the same as the one in [21]. It is satisfied by standard losses such as square error loss and logistic loss.

## 4.2 Theorem statement and discussion

We now present our main theorem on ResNets. Theorem 2 outlines two geometric conditions under which it shows that the critical points of deep ResNets have benign properties.

**Theorem 2.** *Suppose Assumption 4.1 holds. Let*

$$\boldsymbol{\theta}^* := (\boldsymbol{w}^*, \boldsymbol{V}_1^*, \boldsymbol{V}_2^*, \boldsymbol{U}_2^*, \ldots, \boldsymbol{V}_L^*, \boldsymbol{U}_L^*, \boldsymbol{z}^*)$$

*be any twice-differentiable critical point of $\mathfrak{R}(\cdot)$. If*

- $\mathbb{E}_{(x,y)\sim\mathcal{P}}\left[\ell''(f_{\boldsymbol{\theta}^*}(x); y)h_L(x)h_L(x)^T\right]$ *is full-rank; and*
- $\mathrm{col}\left(\left[(\boldsymbol{U}_2^*)^T \quad \cdots \quad (\boldsymbol{U}_L^*)^T\right]\right) \subsetneq \mathbb{R}^{d_x}$,

*then at least one of the following inequalities holds:*

- $\mathfrak{R}(\boldsymbol{\theta}^*) \leq \mathfrak{R}_{\mathrm{lin}}$.
- $\lambda_{\min}(\nabla^2\mathfrak{R}(\boldsymbol{\theta}^*)) < 0$.

The proof of Theorem 2 is deferred to Appendix A. Theorem 2 shows that if the two geometric and linear-algebraic conditions hold, then the risk function value for $f_{\boldsymbol{\theta}^*}$ is at least as good as the best linear predictor, or there is a strict negative eigenvalue of the Hessian at $\boldsymbol{\theta}^*$ so that it is easy to escape from this saddle point. A direct implication of these conditions is that if they continue to hold over the optimization process, then with curvature sensitive algorithms we can find a local minimum no worse than the best linear predictor; notice that our result holds for general losses and data distributions.

As noted earlier, if $L = 1$, our ResNet reduces down to the one considered in [21]. In this case, the second condition is always satisfied because it does not involve the first residual block. In fact, our proof reveals that in the $L = 1$ case, any critical point with $\boldsymbol{w}^* \neq 0$ satisfies $\mathfrak{R}(\boldsymbol{\theta}^*) \leq \mathfrak{R}_{\mathrm{lin}}$ even without the first condition, which recovers the key implication of [21, Theorem 1]. We again emphasize that Theorem 2 extends the previous result.

Theorem 2 also implies something noteworthy about the role of skip-connections in general. Existing results featuring beneficial impacts of skip-connections or parellel shortcut networks on optimization landscapes require direct connection to output [13, 14, 19] or the last hidden layer [21]. The multi-block ResNet we consider in our paper is fundamentally different from other works; the skip-connections connect input to output *through a chain of multiple skip-connections*. Our paper proves that multiple skip-connections (as opposed to direct) can also improve the optimization landscape of neural networks, as was observed empirically [12].

We now discuss the conditions. We call the first condition the *representation coverage condition*, because it requires that the representation $h_L(x)$ by the last residual block "covers" the full space $\mathbb{R}^{d_x}$ so that $\mathbb{E}_{(x,y)\sim\mathcal{P}}\left[\ell''(f_{\boldsymbol{\theta}}(x); y)h_L(x)h_L(x)^T\right]$ is full rank. Especially in cases where $\ell$ is strictly convex, this condition is very mild and likely to hold in most cases.

The second condition is the *parameter coverage condition*. It requires that the subspace spanned by the rows of $\boldsymbol{U}_2^*, \ldots, \boldsymbol{U}_L^*$ is not the full space $\mathbb{R}^{d_x}$. This condition means that the parameters $\boldsymbol{U}_2^*, \ldots, \boldsymbol{U}_L^*$ do *not* cover the full feature space $\mathbb{R}^{d_x}$, so there is some information in the data/representation that this network "misses," which enables us to easily find a direction to improve the parameters.

These conditions stipulate that if the data representation is "rich" enough but the parameters do not cover the full space, then there is always a sufficient room for improvement. We also note that there is an architectural *sufficient condition* $\sum_{l=2}^{L} m_l < d_x$ for our parameter coverage condition to *always* hold, which yields the following noteworthy corollary:

**Corollary 3.** *Suppose Assumption 4.1 holds. For a ResNet $f_{\boldsymbol{\theta}}(\cdot)$ that satisfies $\sum_{l=2}^{L} m_l < d_x$, let $\boldsymbol{\theta}^*$ be a twice-differentiable critical point of $\mathfrak{R}(\cdot)$. Then, the conclusion of Theorem 2 holds as long as $\mathbb{E}_{(x,y)\sim\mathcal{P}}\left[\ell''(f_{\boldsymbol{\theta}^*}(x); y)h_L(x)h_L(x)^T\right]$ is full-rank.*

**Example.** Consider a deep ResNet with very simple residual blocks: $h \mapsto h + \boldsymbol{v}_l\sigma(\boldsymbol{u}_l^T h)$, where $\boldsymbol{v}_l, \boldsymbol{u}_l \in \mathbb{R}^{d_x}$ are vectors and $\sigma$ is ReLU activation. Even this simple architecture is a universal approximator [15]. Notice that Corollary 3 applies to this architecture as long as the depth $L \leq d_x$.

The reader may be wondering what happens if the coverage conditions are not satisfied. In particular, if the parameter coverage condition is not satisfied, i.e., col $\left( \left[ (\boldsymbol{U}_2^*)^T \quad \cdots \quad (\boldsymbol{U}_L^*)^T \right] \right) = \mathbb{R}^{d_x}$, we conjecture that since the parameters already cover the full feature space, the critical point should be of "good" quality. However, we leave a weakening/removal of our geometric conditions to future work.

# 5 Benign properties in near-identity regions of ResNets

This section studies near-identity regions in optimization and generalization aspects, and shows interesting bounds that hold in near-identity regions. We first show an upper bound on the risk value at critical points, and show that the bound is $\mathfrak{R}_{\text{lin}}$ plus a size-independent (i.e., independent of depth and width) constant if the Lipschitz constants of $\Phi_{\boldsymbol{\theta}}^l$'s satisfy $O(1/L)$. We then prove a Rademacher complexity bound on ResNets, and show that the bound also becomes size-independent if $\Phi_{\boldsymbol{\theta}}^l$ is $O(1/L)$-Lipschitz.

## 5.1 Upper bound on the risk value at critical points

Even without the geometric conditions in Section 4, can we prove an upper bound on the risk value of critical points? We prove that for general architectures, the risk value of critical points can be bounded above by $\mathfrak{R}_{\text{lin}}$ plus an additive term. Surprisingly, if each residual block is close to identity, this additive term becomes depth-independent.

In this subsection, the residual parts $\Phi_{\boldsymbol{\theta}}^l(\cdot)$ of ResNet can have any general feedforward architecture:
$$\Phi_{\boldsymbol{\theta}}^l(t) = \phi_{\boldsymbol{z}}^l(t), \quad l = 1, \ldots, L.$$
The collection of all parameters is simply $\boldsymbol{\theta} := (\boldsymbol{w}, \boldsymbol{z})$. We make the following assumption on the functions $\phi_{\boldsymbol{z}}^l : \mathbb{R}^{d_x} \mapsto \mathbb{R}^{d_x}$:

**Assumption 5.1.** *For any $l \in [L]$, the residual part $\phi_{\boldsymbol{z}}^l$ is $\rho_l$-Lipschitz, and $\phi_{\boldsymbol{z}}^l(\boldsymbol{0}) = \boldsymbol{0}$.*

For example, this assumption holds for $\phi_{\boldsymbol{z}}^l(t) = \boldsymbol{V}_l\sigma(\boldsymbol{U}_l t)$, where $\sigma$ is ReLU activation. In this case, $\rho_l$ depends on the spectral norm of $\boldsymbol{V}_l$ and $\boldsymbol{U}_l$.

We also make the following assumption on the loss function $\ell$:

**Assumption 5.2.** *The loss function $\ell(p; y)$ is a convex differentiable function of $p$. We also assume that $\ell(p; y)$ is $\mu$-Lipschitz;, i.e., $|\ell'(p; y)| \leq \mu$ for all $p$.*

Under these assumptions, we prove a bound on the risk value attained at critical points of ResNets.

**Theorem 4.** *Suppose Assumptions 5.1 and 5.2 hold. Let $\boldsymbol{\theta}^*$ be any critical point of $\mathfrak{R}(\cdot)$. Let $\hat{t} \in \mathbb{R}^{d_x}$ be any vector that attains the best linear fit, i.e., $\mathfrak{R}_{\text{lin}} = \mathbb{E}_{(x,y)\sim\mathcal{P}}\left[ \ell(\hat{t}^T x; y) \right]$. Then, for any critical point $\boldsymbol{\theta}^*$ of $\mathfrak{R}(\cdot)$,*
$$\mathfrak{R}(\boldsymbol{\theta}^*) \leq \mathfrak{R}_{\text{lin}} + \mu\|\hat{t}\| \left( \prod_{l=1}^L (1 + \rho_l) - 1 \right) \mathbb{E}_{(x,y)\sim\mathcal{P}}[\|x\|].$$

The proof can be found in Appendix B. Theorem 4 provides an upper bound on $\mathfrak{R}(\boldsymbol{\theta}^*)$ for critical points, without any conditions as in Theorem 2. Of course, depending on the values of constants, the bound could be way above $\mathfrak{R}_{\text{lin}}$. However, if $\rho_l = O(1/L)$, the term $\prod_{l=1}^L (1 + \rho_l)$ is bounded above by a constant, so the additive term in the upper bound becomes size-independent. Furthermore, if $\rho_l = o(1/L)$, the term $\prod_{l=1}^L (1 + \rho_l) \to 1$ as $L \to \infty$, so the additive term in the upper bound diminishes to zero as the network gets deeper. This result indicates that the near-identity region has a good optimization landscape property that any critical point has a risk value that is not too far off from $\mathfrak{R}_{\text{lin}}$.

## 5.2 Radamacher complexity of ResNets

In this subsection, we consider ResNets with the following residual part:
$$\Phi_{\boldsymbol{\theta}}^l(t) = \boldsymbol{V}_l\sigma(\boldsymbol{U}_l t), \quad l = 1, \ldots, L,$$

where $\sigma$ is ReLU activation, $\boldsymbol{V}_l \in \mathbb{R}^{d_x \times d_l}, \boldsymbol{U}_l \in \mathbb{R}^{d_l \times d_x}$. For this architecture, we prove an upper bound on empirical Rademacher complexity that is size-independent in the near-identity region.

Given a set $S = (x_1, \ldots, x_n)$ of $n$ samples, and a class $\mathcal{F}$ of real-valued functions defined on $\mathcal{X}$, the **empirical Rademacher complexity** or **Rademacher averages** of $\mathcal{F}$ restricted to $S$ (denoted as $\mathcal{F}|_S$) is defined as

$$\widehat{\mathscr{R}}_n(\mathcal{F}|_S) = \mathbb{E}_{\epsilon_{1:n}} \left[ \sup_{f \in \mathcal{F}} \frac{1}{n} \sum_{i=1}^{n} \epsilon_i f(x_i) \right],$$

where $\epsilon_i$, $i = 1, \ldots n$, are i.i.d. Rademacher random variables (i.e., Bernoulli coin flips with probability 0.5 and outcome $\pm 1$).

We now state the main result, which proves an upper bound on the Rademacher averages of the class of ResNet functions on a compact domain and norm-bounded parameters.

**Theorem 5.** *Given a set $S = (x_1, \ldots, x_n)$, suppose $\|x_i\| \leq B$ for all $i \in [n]$. Define the function class $\mathcal{F}_L$ of $L$-block ResNet with parameter constraints as:*

$$\mathcal{F}_L := \{f_{\boldsymbol{\theta}} : \mathbb{R}^{d_x} \mapsto \mathbb{R} \mid \|\boldsymbol{w}\| \leq 1, \text{ and } \|\boldsymbol{V}_l\|_{\mathrm{F}}, \|\boldsymbol{U}_l\|_{\mathrm{F}} \leq M_l \text{ for all } l \in [L]\}.$$

*Then, the empirical Rademacher complexity satisfies*

$$\widehat{\mathscr{R}}_n(\mathcal{F}_L|_S) \leq \frac{B \prod_{l=1}^{L}(1 + 2M_l^2)}{\sqrt{n}}.$$

The proof of Theorem 5 is deferred to Appendix C. The proof technique used in Theorem 5 is to "peel off" the blocks: we upper-bound the Rademacher complexity of a $l$-block ResNet with that of a $(l-1)$-block ResNet multiplied by $1 + 2M_l^2$. Consider a fully-connected network $x \mapsto \boldsymbol{W}_L \sigma(\boldsymbol{W}_{L-1} \cdots \sigma(\boldsymbol{W}_1 x) \cdots)$, where $\boldsymbol{W}_l$'s are weight matrices and $\sigma$ is ReLU activation. The same "peeling off" technique was used in [16], which showed a bound of $O\left(B \cdot 2^L \prod_{l=1}^{L} C_l / \sqrt{n}\right)$, where $C_l$ is the Frobenius norm bound of $\boldsymbol{W}_l$. As we can see, this bound has an exponential dependence on depth $L$, which is difficult to remove. Other results [2, 17] reduced the dependence down to polynomial, but it wasn't until the work by Golowich et al. [4] that a size-independent bound became known. However, their size-independent bound has worse dependence on $n$ ($O(1/n^{1/4})$) than other bounds ($O(1/\sqrt{n})$).

In contrast, Theorem 5 shows that for ResNets, the upper bound easily becomes *size-independent* as long as $M_l = O(1/\sqrt{L})$, which is surprising. Of course, for fully-connected networks, the upper bound above can also be made size-independent by forcing $C_l \leq 1/2$ for all $l \in [L]$. However, in this case, the network becomes **trivial**, meaning that the output has to be very close to zero for any input $x$. In case of ResNets, the difference is that the bound can be made size-independent **even for non-trivial networks**.

## 6   Conclusion

We investigated the question whether local minima of risk function of a *deep* ResNet are better than linear predictors. We showed two motivating examples showing 1) the advantage of ResNets over fully-connected networks, and 2) difficulty in analysis of deep ResNets. Then, we showed that under geometric conditions, any critical point of the risk function of a deep ResNet has benign properties that it is either better than linear predictors or the Hessian at the critical point has a strict negative eigenvalue. We supplement the result by showing size-independent upper bounds on the risk value of critical points as well as empirical Rademacher complexity for near-identity regions of deep ResNets. We hope that this work becomes a stepping stone on deeper understanding of ResNets.

**Acknowledgments**

All the authors acknowledge support from DARPA Lagrange. Chulhee Yun also thanks Korea Foundation for Advanced Studies for their support. Suvrit Sra also acknowledges support from an NSF-CAREER grant and an Amazon Research Award.

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
