[Supplementary Material · main_skipconn-11-15.pdf]

# A   Proof of Theorem 2

Before we begin the proof, let us introduce more notation. Since we only consider a single critical point, for simplicity of notation we denote the critical point as $\boldsymbol{\theta} = (\boldsymbol{w}, \boldsymbol{V}_1, \boldsymbol{V}_2, \boldsymbol{U}_2, \ldots, \boldsymbol{V}_L, \boldsymbol{U}_L, \boldsymbol{z})$, without $*$. For $l \in [2 : L]$, let $J_l(x) := \nabla \phi_{\boldsymbol{z}}^l(\boldsymbol{U}_l h_{l-1}(x)) \in \mathbb{R}^{n_l \times m_l}$, i.e., $J^l(x)$ is the Jacobian matrix of $\phi_{\boldsymbol{z}}^l(\cdot)$ evaluated at $\boldsymbol{U}_l h_{l-1}(x)$, whenever it exists. Also, let $\mathcal{U} := \operatorname{col}\left(\begin{bmatrix} \boldsymbol{U}_2^T & \cdots & \boldsymbol{U}_L^T \end{bmatrix}\right) \subsetneq \mathbb{R}^{d_x}$.

The proof is divided into two cases: 1) if $\boldsymbol{w} \notin \mathcal{U}$, and 2) if $\boldsymbol{w} \in \mathcal{U}$. For Case 1, we will show that $\mathfrak{R}(\boldsymbol{\theta}^*) \leq \mathfrak{R}_{\mathrm{lin}}$; we also note that our representation coverage condition $\operatorname{rank}(\mathbb{E}_{(x,y)\sim\mathcal{P}}\left[\ell''(f_{\boldsymbol{\theta}}(x); y) h_L(x) h_L(x)^T\right]) = d_x$ is not required for Case 1. For Case 2, we will show that at least one of $\mathfrak{R}(\boldsymbol{\theta}^*) \leq \mathfrak{R}_{\mathrm{lin}}$ or $\lambda_{\min}(\nabla^2 \mathfrak{R}(\boldsymbol{\theta}^*)) < 0$ has to hold.

**Case 1: If $w \notin \mathcal{U}$.**  From standard matrix calculus, we can calculate the partial derivatives of $\mathfrak{R}$ with respect to $\boldsymbol{w}$ and $\boldsymbol{V}_l$'s. Since $\boldsymbol{\theta}$ is a critical point we have

$$\frac{\partial \mathfrak{R}}{\partial \boldsymbol{w}}(\boldsymbol{\theta}) = \mathbb{E}\left[\ell'(f_{\boldsymbol{\theta}}(x); y) h_L(x)\right] = \boldsymbol{0},$$

$$\frac{\partial \mathfrak{R}}{\partial \boldsymbol{V}_l}(\boldsymbol{\theta}) = \mathbb{E}\left[\ell'(f_{\boldsymbol{\theta}}(x); y) \prod_{k=l+1}^{L} (I + \boldsymbol{U}_k^T J_k(x)^T \boldsymbol{V}_k^T) \boldsymbol{w} \phi_{\boldsymbol{z}}^l (U_l h_{l-1}(x))^T\right] = \boldsymbol{0}, \quad l = 2, \ldots, L,$$

$$\frac{\partial \mathfrak{R}}{\partial \boldsymbol{V}_1}(\boldsymbol{\theta}) = \mathbb{E}\left[\ell'(f_{\boldsymbol{\theta}}(x); y) \prod_{k=2}^{L} (I + \boldsymbol{U}_k^T J_k(x)^T \boldsymbol{V}_k^T) \boldsymbol{w} \phi_{\boldsymbol{z}}^1 (x)^T\right] = \boldsymbol{0}.$$

For $\boldsymbol{V}_2, \ldots, \boldsymbol{V}_L$, note that we can arrange terms and express the partial derivatives as

$$\frac{\partial \mathfrak{R}}{\partial \boldsymbol{V}_l}(\boldsymbol{\theta}) = \boldsymbol{w}\mathbb{E}\left[\ell'(f_{\boldsymbol{\theta}}(x); y) \phi_{\boldsymbol{z}}^l (U_l h_{l-1}(x))\right]^T + \sum_{k=l+1}^{L} \boldsymbol{U}_k^T E_k = \boldsymbol{0}, \tag{1}$$

where $E_k \in \mathbb{R}^{m_l \times n_l}$ are appropriately defined matrices. Note that any column of $\sum_{k=l+1}^{L} \boldsymbol{U}_k^T E_k$ is in $\mathcal{U}$. Since $\boldsymbol{w} \notin \mathcal{U}$, the sum being zero (1) implies that $\mathbb{E}\left[\ell'(f_{\boldsymbol{\theta}}(x); y) \phi_{\boldsymbol{z}}^l (U_l h_{l-1}(x))\right] = \boldsymbol{0}$ (because $\boldsymbol{w} \notin \mathcal{U}$ already implies that $\boldsymbol{w} \neq 0$), for all $l \in [2 : L]$. Similarly, we have $\mathbb{E}\left[\ell'(f_{\boldsymbol{\theta}}(x); y) \phi_{\boldsymbol{z}}^1 (x)\right] = \boldsymbol{0}$.

Now, from $\mathbb{E}\left[\ell'(f_{\boldsymbol{\theta}}(x); y) h_L(x)\right] = \boldsymbol{0}$,

$$\begin{aligned}
\boldsymbol{0} &= \mathbb{E}\left[\ell'(f_{\boldsymbol{\theta}}(x); y) h_L(x)\right] \\
&= \mathbb{E}\left[\ell'(f_{\boldsymbol{\theta}}(x); y)\left(h_{L-1}(x) + \boldsymbol{V}_L \phi_{\boldsymbol{z}}^L (U_L h_{L-1}(x))\right)\right] \\
&= \mathbb{E}\left[\ell'(f_{\boldsymbol{\theta}}(x); y) h_{L-1}(x)\right] + \boldsymbol{V}_L \mathbb{E}\left[\ell'(f_{\boldsymbol{\theta}}(x); y) \phi_{\boldsymbol{z}}^L (U_L h_{L-1}(x))\right] \\
&= \mathbb{E}\left[\ell'(f_{\boldsymbol{\theta}}(x); y) h_{L-1}(x)\right] = \cdots = \mathbb{E}\left[\ell'(f_{\boldsymbol{\theta}}(x); y) x\right].
\end{aligned}$$

Recall that by convexity, $\ell(p; y) - \ell(q; y) \leq \ell'(p; y)(p - q)$. Now for any $t \in \mathbb{R}^{d_x}$, we can apply this inequality for $p = f_{\boldsymbol{\theta}}(x) = \boldsymbol{w}^T h_L(x)$ and $q = t^T x$:

$$\begin{aligned}
\mathbb{E}\left[\ell(f_{\boldsymbol{\theta}}(x); y)\right] - \mathbb{E}\left[\ell(t^T x; y)\right] &\leq \mathbb{E}\left[\ell'(f_{\boldsymbol{\theta}}(x); y)(\boldsymbol{w}^T h_L(x) - t^T x)\right] \\
&= \boldsymbol{w}^T \mathbb{E}\left[\ell'(f_{\boldsymbol{\theta}}(x); y) h_L(x)\right] - t^T \mathbb{E}\left[\ell'(f_{\boldsymbol{\theta}}(x); y) x\right] = 0.
\end{aligned}$$

Thus, $\mathbb{E}\left[\ell(f_{\boldsymbol{\theta}}(x); y)\right] \leq \mathbb{E}\left[\ell(t^T x; y)\right]$ for all $t$, so taking infimum over $t$ gives $\mathfrak{R}(\boldsymbol{\theta}^*) \leq \mathfrak{R}_{\mathrm{lin}}$.

**Case 2: If $w \in \mathcal{U}$.**  For this case, we will consider the Hessian of $\mathfrak{R}$ with respect to $\boldsymbol{w}$ and $\boldsymbol{V}_l$, for each $l \in [L]$. We will show that if $\mathbb{E}\left[\ell'(f_{\boldsymbol{\theta}}(x); y) \phi_{\boldsymbol{z}}^l (U_l h_{l-1}(x))\right] \neq \boldsymbol{0}$, then $\lambda_{\min}(\nabla^2 \mathfrak{R}(\boldsymbol{\theta})) < 0$. This implies that if $\mathbb{E}\left[\ell'(f_{\boldsymbol{\theta}}(x); y) \phi_{\boldsymbol{z}}^l (U_l h_{l-1}(x))\right] = \boldsymbol{0}$ for all $l \in [L]$, then by the same argument as in Case 1 we have $\mathfrak{R}(\boldsymbol{\theta}^*) \leq \mathfrak{R}_{\mathrm{lin}}$; otherwise, we have $\lambda_{\min}(\nabla^2 \mathfrak{R}(\boldsymbol{\theta})) < 0$.

Because $\boldsymbol{\theta}$ is a twice-differentiable critical point of $\mathfrak{R}(\cdot)$, if we apply perturbation $\boldsymbol{\delta}$ to $\boldsymbol{\theta}$ and do Taylor expansions, what we get is

$$\mathfrak{R}(\boldsymbol{\theta} + \boldsymbol{\delta}) = \mathfrak{R}(\boldsymbol{\theta}) + \tfrac{1}{2}\boldsymbol{\delta}^T \nabla^2 \mathfrak{R}(\boldsymbol{\theta})\boldsymbol{\delta} + o(\|\boldsymbol{\delta}\|^2). \tag{2}$$

So, if we apply a particular form of perturbation $\boldsymbol{\delta}$, calculate $\Re(\boldsymbol{\theta} + \boldsymbol{\delta})$, and then show that the sum of all second-order perturbation terms are negative for such a $\boldsymbol{\delta}$, it is equivalent to showing $\frac{1}{2}\boldsymbol{\delta}^T\nabla^2\Re(\boldsymbol{\theta})\boldsymbol{\delta} < 0$, hence $\lambda_{\min}(\nabla^2\Re(\boldsymbol{\theta})) < 0$.

Now fix any $l \in [2:L]$, and consider perturbing $\boldsymbol{w}$ by $\boldsymbol{\epsilon}$ and $\boldsymbol{V}_l$ by $\boldsymbol{\Delta}$, while leaving all other parameters unchanged. We will choose $\boldsymbol{\Delta} = \alpha\beta^T$, where $\alpha \in \mathbb{R}^{d_x}$ is chosen from $\alpha \in \mathcal{U}^\perp$, the orthogonal complement of $\mathcal{U}$, and $\beta \in \mathbb{R}^{n_l}$ will be chosen later. We will now compute $\Re(\boldsymbol{\theta} + \boldsymbol{\delta})$ directly from the network architecture. The residual block output $h_1(x), \ldots, h_{l-1}(x)$ stays invariant after perturbation because their parameters didn't change. For $l$-th residual block, the output after perturbation, denoted as $\tilde{h}_l(x)$, becomes

$$\tilde{h}_l(x) = h_l(x) + \boldsymbol{\Delta}\phi_{\boldsymbol{z}}^l(\boldsymbol{U}_l h_{l-1}(x)).$$

The next residual block output is

$$\begin{aligned}
\tilde{h}_{l+1}(x) &= \tilde{h}_l(x) + \boldsymbol{V}_{l+1}\phi_{\boldsymbol{z}}^{l+1}(\boldsymbol{U}_{l+1}\tilde{h}_l(x)) \\
&= h_l(x) + \boldsymbol{\Delta}\phi_{\boldsymbol{z}}^l(\boldsymbol{U}_l h_{l-1}(x)) + \boldsymbol{V}_{l+1}\phi_{\boldsymbol{z}}^{l+1}\big(\boldsymbol{U}_{l+1}h_l(x) + \boldsymbol{U}_{l+1}\boldsymbol{\Delta}\phi_{\boldsymbol{z}}^l(\boldsymbol{U}_l h_{l-1}(x))\big) \\
&\overset{(a)}{=} h_l(x) + \boldsymbol{\Delta}\phi_{\boldsymbol{z}}^l(\boldsymbol{U}_l h_{l-1}(x)) + \boldsymbol{V}_{l+1}\phi_{\boldsymbol{z}}^{l+1}(\boldsymbol{U}_{l+1}h_l(x)) \\
&= h_{l+1}(x) + \boldsymbol{\Delta}\phi_{\boldsymbol{z}}^l(\boldsymbol{U}_l h_{l-1}(x)),
\end{aligned}$$

where (a) used the fact that $\boldsymbol{U}_{l+1}\boldsymbol{\Delta} = \boldsymbol{U}_{l+1}\alpha\beta^T = 0$ because $\alpha \in \mathcal{U}^\perp$. We can propagate this up to $\tilde{h}_L(x)$ and similarly show $\tilde{h}_L(x) = h_L(x) + \boldsymbol{\Delta}\phi_{\boldsymbol{z}}^l(\boldsymbol{U}_l h_{l-1}(x))$. Using this, the network output after perturbation, denoted as $f_{\boldsymbol{\theta}+\boldsymbol{\delta}}(\cdot)$, is

$$\begin{aligned}
f_{\boldsymbol{\theta}+\boldsymbol{\delta}}(x) &= (\boldsymbol{w} + \boldsymbol{\epsilon})^T\big(h_L(x) + \boldsymbol{\Delta}\phi_{\boldsymbol{z}}^l(\boldsymbol{U}_l h_{l-1}(x))\big) \\
&= f_{\boldsymbol{\theta}}(x) + \boldsymbol{\epsilon}^T h_L(x) + \boldsymbol{w}^T\boldsymbol{\Delta}\phi_{\boldsymbol{z}}^l(\boldsymbol{U}_l h_{l-1}(x)) + \boldsymbol{\epsilon}^T\boldsymbol{\Delta}\phi_{\boldsymbol{z}}^l(\boldsymbol{U}_l h_{l-1}(x)) \\
&\overset{(b)}{=} f_{\boldsymbol{\theta}}(x) + \boldsymbol{\epsilon}^T h_L(x) + \boldsymbol{\epsilon}^T\boldsymbol{\Delta}\phi_{\boldsymbol{z}}^l(\boldsymbol{U}_l h_{l-1}(x)),
\end{aligned}$$

where (b) used $\boldsymbol{w}^T\boldsymbol{\Delta} = \boldsymbol{w}^T\alpha\beta^T = 0$ because $\boldsymbol{w} \in \mathcal{U}$ and $\alpha \in \mathcal{U}^\perp$. Using this, the risk function value after perturbation is

$$\begin{aligned}
\Re(\boldsymbol{\theta} + \boldsymbol{\delta}) &= \mathbb{E}\left[\ell(f_{\boldsymbol{\theta}+\boldsymbol{\delta}}(x); y)\right] \\
&= \mathbb{E}\left[\ell(f_{\boldsymbol{\theta}}(x) + \boldsymbol{\epsilon}^T h_L(x) + \boldsymbol{\epsilon}^T\boldsymbol{\Delta}\phi_{\boldsymbol{z}}^l(\boldsymbol{U}_l h_{l-1}(x)); y)\right] \\
&\overset{(c)}{=} \mathbb{E}\Big[\ell(f_{\boldsymbol{\theta}}(x); y) + \ell'(f_{\boldsymbol{\theta}}(x); y)\big(\boldsymbol{\epsilon}^T h_L(x) + \boldsymbol{\epsilon}^T\boldsymbol{\Delta}\phi_{\boldsymbol{z}}^l(\boldsymbol{U}_l h_{l-1}(x))\big) \\
&\qquad + \tfrac{1}{2}\ell''(f_{\boldsymbol{\theta}}(x); y)\big(\boldsymbol{\epsilon}^T h_L(x)\big)^2 + o(\|\boldsymbol{\delta}\|^2)\Big] \\
&\overset{(d)}{=} \Re(\boldsymbol{\theta}) + \mathbb{E}\left[\ell'(f_{\boldsymbol{\theta}}(x); y)\boldsymbol{\epsilon}^T\boldsymbol{\Delta}\phi_{\boldsymbol{z}}^l(\boldsymbol{U}_l h_{l-1}(x)) + \tfrac{1}{2}\ell''(f_{\boldsymbol{\theta}}(x); y)\big(\boldsymbol{\epsilon}^T h_L(x)\big)^2\right] + o(\|\boldsymbol{\delta}\|^2),
\end{aligned}$$

where (c) used Taylor expansion of $\ell(\cdot; y)$ and (d) used that $\mathbb{E}[\ell'(f_{\boldsymbol{\theta}}(x); y)h_L(x)] = \frac{\partial\Re}{\partial\boldsymbol{w}}(\boldsymbol{\theta}) = \boldsymbol{0}$. Comparing with the expansion (2), the second term in the RHS corresponds to the second-order perturbation $\frac{1}{2}\boldsymbol{\delta}^T\nabla^2\Re(\boldsymbol{\theta})\boldsymbol{\delta}$.

Now note that

$$\begin{aligned}
&\mathbb{E}\left[\ell'(f_{\boldsymbol{\theta}}(x); y)\boldsymbol{\epsilon}^T\boldsymbol{\Delta}\phi_{\boldsymbol{z}}^l(\boldsymbol{U}_l h_{l-1}(x)) + \tfrac{1}{2}\ell''(f_{\boldsymbol{\theta}}(x); y)\big(\boldsymbol{\epsilon}^T h_L(x)\big)^2\right] \\
=&\boldsymbol{\epsilon}^T\boldsymbol{\Delta}\mathbb{E}\left[\ell'(f_{\boldsymbol{\theta}}(x); y)\phi_{\boldsymbol{z}}^l(\boldsymbol{U}_l h_{l-1}(x))\right] + \tfrac{1}{2}\boldsymbol{\epsilon}^T\mathbb{E}\left[\ell''(f_{\boldsymbol{\theta}}(x); y)h_L(x)h_L(x)^T\right]\boldsymbol{\epsilon}.
\end{aligned}$$

Let $A := \mathbb{E}[\ell''(f_{\boldsymbol{\theta}}(x); y)h_L(x)h_L(x)^T]$ and $b := \mathbb{E}\left[\ell'(f_{\boldsymbol{\theta}}(x); y)\phi_{\boldsymbol{z}}^l(\boldsymbol{U}_l h_{l-1}(x))\right]$ for simplicity. By the representation coverage condition of the theorem $A$ is full-rank, hence invertible. We can choose $\boldsymbol{\epsilon} = -A^{-1}\boldsymbol{\Delta}b$ to minimize the expression above, then the minimum value we get is $-\frac{1}{2}b^T\boldsymbol{\Delta}^T A^{-1}\boldsymbol{\Delta}b$.

First, note that $A$ is positive definite, and so is $A^{-1}$. If $b \neq 0$, we can choose $\beta = b$, so $\boldsymbol{\Delta}b = \alpha\beta^T b = \|b\|^2\alpha \neq \boldsymbol{0}$, so $-\frac{1}{2}b^T\boldsymbol{\Delta}^T A^{-1}\boldsymbol{\Delta}b < 0$. This proves that $\lambda_{\min}(\nabla^2\Re(\boldsymbol{\theta})) < 0$ if $\mathbb{E}\left[\ell'(f_{\boldsymbol{\theta}}(x); y)\phi_{\boldsymbol{z}}^l(\boldsymbol{U}_l h_{l-1}(x))\right] \neq 0$, as desired.

The case when $l = 1$ can be done similarly, by perturbing $\boldsymbol{w}$ and $\boldsymbol{V}_1$. This finishes the proof.

# B  Proof of Theorem 4

Since we only consider a single critical point, we denote the critical point as $\boldsymbol{\theta} = (\boldsymbol{w}, \boldsymbol{z})$, without $*$. By the same argument as in Case 1 of Proof of Theorem 2, we can use convexity of $\ell$ to get the following bound:

$$
\begin{aligned}
\mathbb{E}\left[\ell(f_{\boldsymbol{\theta}}(x); y)\right] - \mathbb{E}\left[\ell(\hat{t}^T x; y)\right] &\leq \mathbb{E}\left[\ell'(f_{\boldsymbol{\theta}}(x); y)(\boldsymbol{w}^T h_L(x) - \hat{t}^T x)\right] \\
&= (\boldsymbol{w} - \hat{t})^T \mathbb{E}\left[\ell'(f_{\boldsymbol{\theta}}(x); y) h_L(x)\right] + \hat{t}^T \mathbb{E}\left[\ell'(f_{\boldsymbol{\theta}}(x); y)(h_L(x) - x)\right] \\
&\stackrel{(a)}{=} \hat{t}^T \mathbb{E}\left[\ell'(f_{\boldsymbol{\theta}}(x); y) \sum_{l=1}^{L} \phi_{\boldsymbol{z}}^l(h_{l-1}(x))\right] \\
&\leq \mu \|\hat{t}\| \sum_{l=1}^{L} \mathbb{E}\left[\|\phi_{\boldsymbol{z}}^l(h_{l-1}(x))\|\right],
\end{aligned}
$$

where (a) used the fact that $\mathbb{E}\left[\ell'(f_{\boldsymbol{\theta}}(x); y) h_L(x)\right] = \frac{\partial \mathfrak{R}}{\partial \boldsymbol{w}} = 0$. Now, for any fixed $l \in [L]$, using Assumption 5.1 we have

$$
\begin{aligned}
\|\phi_{\boldsymbol{z}}^l(h_{l-1}(x))\| &\leq \rho_l \|h_{l-1}(x))\| \\
&\leq \rho_l(\|h_{l-2}(x)\| + \|\phi_{\boldsymbol{z}}^{l-1}(h_{l-2}(x))\|) \\
&\leq \rho_l(1 + \rho_{l-1})\|h_{l-2}(x)\| \\
&\leq \cdots \leq \rho_l \prod_{k=1}^{l-1}(1 + \rho_k)\|x\|.
\end{aligned}
$$

Substituting this bound to the one above, we get

$$
\mathfrak{R}(\boldsymbol{\theta}) - \mathfrak{R}_{\text{lin}} \leq \mu \|\hat{t}\| \mathbb{E}\left[\|x\|\right] \sum_{l=1}^{L} \rho_l \prod_{k=1}^{l-1}(1 + \rho_k) = \mu \|\hat{t}\| \mathbb{E}\left[\|x\|\right] \left(\prod_{k=1}^{L}(1 + \rho_k) - 1\right).
$$

# C  Proof of Theorem 5

First, we collect the symbols used in this section. Given a real number $p$, define $[p]_+ := \max\{p, 0\}$ and $[p]_- := \max\{-p, 0\}$. Notice that $|p| = [p]_+ + [p]_-$. Recall that given a vector $x$, let $\|x\|$ denotes its Euclidean norm. Recall also that given a matrix $M$, let $\|M\|$ denote its spectral norm, and $\|M\|_{\text{F}}$ denote its Frobenius norm.

The proof is done by a simple induction argument using the "peeling-off" technique used for Rademacher complexity bounds for neural networks. Before we start, let us define the function class of hidden layer representations, for $0 \leq l \leq L$:

$$
\mathcal{H}_l := \{h_l : \mathbb{R}^{d_x} \mapsto \mathbb{R}^{d_x} \mid \|\boldsymbol{V}_j\|_{\text{F}}, \|\boldsymbol{U}_j\|_{\text{F}} \leq M_j \text{ for all } j \in [l]\},
$$

defined with the same bounds as used in $\mathcal{F}_L$. Note that $\mathcal{H}_0$ is a singleton with the identity mapping $x \mapsto x$. Also, define $\mathcal{F}_l$ to be the class of functions represented by a $l$-block ResNet ($0 \leq l \leq L$):

$$
\mathcal{F}_l := \{x \mapsto w^T h_l(x) \mid \|w\| \leq 1, h_l \in \mathcal{H}_l\}.
$$

Note that if $l = L$, this recovers the definition of $\mathcal{F}_L$ in the theorem statement. Since

$$
\mathcal{F}_0 := \{x \mapsto w^T x \mid \|w\| \leq 1\},
$$

it is well-known that $\widehat{\mathscr{R}}_n(\mathcal{F}_0|_S) \leq \frac{B}{\sqrt{n}}$. The rest of the proof is done by proving the following:

$$
\widehat{\mathscr{R}}_n(\mathcal{F}_l|_S) \leq (1 + 2M_l^2)\widehat{\mathscr{R}}_n(\mathcal{F}_{l-1}|_S),
$$

for $l \in [L]$.

Fix any $l \in [L]$. Then, by the definition of Rademacher complexity,

$$
n\widehat{\mathscr{R}}_n(\mathcal{F}_l|_S) = \mathbb{E}_{\epsilon_{1:n}}\left[\sup_{\substack{\|w\|\leq 1, \\ h_l \in \mathcal{H}_l}} \sum_{i=1}^n \epsilon_i w^T h_l(x_i)\right]
$$

$$
= \mathbb{E}_{\epsilon_{1:n}}\left[\sup_{\substack{\|w\|\leq 1, \\ h_{l-1}\in\mathcal{H}_{l-1}}} \sup_{\substack{\|\boldsymbol{V}_l\|_{\mathrm{F}}\leq M_l \\ \|\boldsymbol{U}_l\|_{\mathrm{F}}\leq M_l}} \sum_{i=1}^n \epsilon_i w^T(h_{l-1}(x_i) + \boldsymbol{V}_l\sigma(\boldsymbol{U}_l h_{l-1}(x_i)))\right]
$$

$$
\leq \mathbb{E}_{\epsilon_{1:n}}\left[\sup_{\substack{\|w\|\leq 1, \\ h_{l-1}\in\mathcal{H}_{l-1}}} \sum_{i=1}^n \epsilon_i w^T h_{l-1}(x_i)\right] + \underbrace{\mathbb{E}_{\epsilon_{1:n}}\left[\sup_{\substack{\|w\|\leq 1, \\ h_{l-1}\in\mathcal{H}_{l-1}}} \sup_{\substack{\|\boldsymbol{V}_l\|_{\mathrm{F}}\leq M_l \\ \|\boldsymbol{U}_l\|_{\mathrm{F}}\leq M_l}} \sum_{i=1}^n \epsilon_i w^T \boldsymbol{V}_l\sigma(\boldsymbol{U}_l h_{l-1}(x_i))\right]}_{=:\mathscr{A}}.
$$

The first term in RHS is $n\widehat{\mathscr{R}}_n(\mathcal{F}_{l-1}|_S)$ by definition. It is left to show an upper bound for the second term in RHS, which we will call $\mathscr{A}$.

First, because $\|w\| \leq 1$ and $\|\boldsymbol{V}_l\| \leq \|\boldsymbol{V}_l\|_{\mathrm{F}} \leq M_l$, we have $\|\boldsymbol{V}_l^T w\| \leq M_l$. So, by using dual norm,

$$
\mathscr{A} = \mathbb{E}\left[\sup_{\substack{\|v\|\leq M_l, \\ \|\boldsymbol{U}_l\|_{\mathrm{F}}\leq M_l \\ h_{l-1}\in\mathcal{H}_{l-1}}} v^T \sum_{i=1}^n \epsilon_i\sigma(\boldsymbol{U}_l h_{l-1}(x_i))\right] = M_l\mathbb{E}\left[\sup_{\substack{\|\boldsymbol{U}_l\|_{\mathrm{F}}\leq M_l, \\ h_{l-1}\in\mathcal{H}_{l-1}}} \left\|\sum_{i=1}^n \epsilon_i\sigma(\boldsymbol{U}_l h_{l-1}(x_i))\right\|\right].
$$

Let $u_1^T, u_2^T, \ldots, u_k^T$ be the rows of $\boldsymbol{U}_l$. Then, by positive homogeneity of ReLU $\sigma$, we have

$$
\left\|\sum_{i=1}^n \epsilon_i\sigma(\boldsymbol{U}_l h_{l-1}(x_i))\right\|^2 = \sum_{j=1}^k \|u_j\|^2\left(\sum_{i=1}^n \epsilon_i\sigma\left(\frac{u_j^T h_{l-1}(x_i)}{\|u_j\|}\right)\right)^2.
$$

The supremum of this quantity over $u_1, \ldots, u_k$ under the constraint that $\|\boldsymbol{U}_l\|_{\mathrm{F}}^2 = \sum_{j=1}^k \|u_j\|^2 \leq M_l^2$ is attained when $\|u_j\| = M_l$ for some $j$ and $\|u_{j'}\| = 0$ for all other $j' \neq j$. This means that

$$
\frac{\mathscr{A}}{M_l} = \mathbb{E}\left[\sup_{\substack{\|\boldsymbol{U}_l\|_{\mathrm{F}}\leq M_l, \\ h_{l-1}\in\mathcal{H}_{l-1}}} \left\|\sum_{i=1}^n \epsilon_i\sigma(\boldsymbol{U}_l h_{l-1}(x_i))\right\|\right] = \mathbb{E}\left[\sup_{\substack{\|u\|\leq M_l, \\ h_{l-1}\in\mathcal{H}_{l-1}}} \left|\sum_{i=1}^n \epsilon_i\sigma(u^T h_{l-1}(x_i))\right|\right]
$$

$$
= \mathbb{E}\left[\sup_{\substack{\|u\|\leq M_l, \\ h_{l-1}\in\mathcal{H}_{l-1}}} \left[\sum_{i=1}^n \epsilon_i\sigma(u^T h_{l-1}(x_i))\right]_+ + \left[\sum_{i=1}^n \epsilon_i\sigma(u^T h_{l-1}(x_i))\right]_-\right]
$$

$$
\leq \mathbb{E}\left[\sup_{\substack{\|u\|\leq M_l, \\ h_{l-1}\in\mathcal{H}_{l-1}}} \left[\sum_{i=1}^n \epsilon_i\sigma(u^T h_{l-1}(x_i))\right]_+\right] + \mathbb{E}\left[\sup_{\substack{\|u\|\leq M_l, \\ h_{l-1}\in\mathcal{H}_{l-1}}} \left[\sum_{i=1}^n \epsilon_i\sigma(u^T h_{l-1}(x_i))\right]_-\right]
$$

$$
\overset{(a)}{=} 2\mathbb{E}\left[\sup_{\substack{\|u\|\leq M_l, \\ h_{l-1}\in\mathcal{H}_{l-1}}} \left[\sum_{i=1}^n \epsilon_i\sigma(u^T h_{l-1}(x_i))\right]_+\right] \overset{(b)}{=} 2\mathbb{E}\left[\left[\sup_{\substack{\|u\|\leq M_l, \\ h_{l-1}\in\mathcal{H}_{l-1}}} \sum_{i=1}^n \epsilon_i\sigma(u^T h_{l-1}(x_i))\right]_+\right]
$$

$$
\overset{(c)}{=} 2\mathbb{E}\left[\sup_{\substack{\|u\|\leq M_l, \\ h_{l-1}\in\mathcal{H}_{l-1}}} \sum_{i=1}^n \epsilon_i\sigma(u^T h_{l-1}(x_i))\right] \overset{(d)}{\leq} 2\mathbb{E}\left[\sup_{\substack{\|u\|\leq M_l, \\ h_{l-1}\in\mathcal{H}_{l-1}}} \sum_{i=1}^n \epsilon_i u^T h_{l-1}(x_i)\right],
$$

where equality (a) is due to symmetry of Rademacher random variables and (b) uses $\sup[t]_+ = [\sup t]_+$. Equality (c) uses the fact that the supremum is nonnegative, because setting $u = 0$

already gives $\sum_{i=1}^{n} \epsilon_i \sigma(u^T h_{l-1}(x_i)) = 0$. Inequality (d) uses contraction property of Rademacher complexity.

Lastly, one can notice that

$$\mathbb{E}\left[\sup_{\substack{\|u\| \leq M_l, \\ h_{l-1} \in \mathcal{H}_{l-1}}} \sum_{i=1}^{n} \epsilon_i u^T h_{l-1}(x_i)\right] = M_l \mathbb{E}\left[\sup_{\substack{\|w\| \leq 1, \\ h_{l-1} \in \mathcal{H}_{l-1}}} \sum_{i=1}^{n} \epsilon_i w^T h_{l-1}(x_i)\right] = M_l n \widehat{\mathscr{R}}_n(\mathcal{F}_{l-1}|_S).$$

This establishes

$$\mathscr{A} \leq 2M_l^2 n \widehat{\mathscr{R}}_n(\mathcal{F}_{l-1}|_S),$$

which leads to the conclusion that

$$\widehat{\mathscr{R}}_n(\mathcal{F}_l|_S) \leq (1 + 2M_l^2)\widehat{\mathscr{R}}_n(\mathcal{F}_{l-1}|_S),$$

as desired.