[Reviews · NeurIPS 2019]

Reviewer 1



This paper tries to answer the question: can the results from [Shamir, 2018] be extended to multi-block ResNets? The authors formally characterize that under condition Line 206 and Line 207, ANY local minimum is either "good" (meaning it is better than linear predictors), or the minimum eigenvalue of its Hessian is strictly negative. These results are new as far as I know. My main concern is regarding the usefulness of the result. In particular, the assumption in Line 207 seems strong compared to the results in [Shamir, 2018]. Also, the result in Section 3.2 seems like a trivial application of [Shamir, 2018]. Overall the paper is easy to follow. I feel there is room for improvements in some sections. Some of them are listed below.

Reviewer 2



The topic is timely and the presentation is crisp and clear. The results shed some light to ResNets, and this is a good technical advance.

Reviewer 3



The paper theoretically investigates deep residual networks, extending a result by Shamir (citation [20] in the paper). First, two motivating examples are given, showing that a linear predictor might outperform all the local minima of a fully connected network, while at the same time a ResNet has strictly better minima. The second example shows that gradually adding residual blocks won't necessarily improve local minima monotonically. Then, the authors show that the critical points of a deep ResNets either have better objective value than a linear predictor, or that they have a strictly negative eigenvalue (hence they are not a local minimum). This is done under a 'representation coverage condition' and another 'parameter coverage condition', where the latter seems rather limiting. Lastly, analysis of deep ResNets is given in the 'near-identity' regions, where the residual block output is small. An upper bound on the objective of such critical points is given, as well as a bound on the Rademacher-complexity of the ResNet. It seems that the near-identity assumption limits these results, although these might be arguably a reasonable assumption given known theory about this setting. My main concern regarding this paper is that proofs applying to deep Resnets, which are mainly based on a recursive analysis of each residual block, do not seem to provide deeper insights into the nature of ResNets, as far as I can tell. Therefore, the techniques used might not result in future improvement based on them. Moreover, the second assumption made in theorem 2 seems rather limiting, as to the best of my knowledge, there's no reason why it should hold for a 'general' local minimum when the condition in corollary 3 isn't met, which seems like a rather strong assumption on the architecture. The examples provided make understanding and reading the paper easier, and provide motivation for what follows, but are nevertheless simple, tailored examples which I do not view as standalone results (nor is it their purpose in the paper). Additionally, section 5 strongly relies on the near-identity assumption which also limits the results in it. While I found these results interesting, they seem in my humble opinion like incremental improvements. Lastly, in line 119 it is claimed that augmenting one dimension per layer extends the results for networks with bias terms, however I'm rather convinced that this trick only works for the first layer. Setting the augmented dimension in the data to all ones simulates bias terms for the first layer, but this does not extend further. Did I miss anything? Can you please elaborate on this trick? Overall, although the paper is clear and easy to follow, and as much as it is exciting to see theoretical works analyzing deep ResNets, this alone does not merit acceptance as the collection of results presented in this paper seem incremental for the most part. Minor comments: Line 78: examples shows -> example shows Line 207: I'm not sure where and if was col(...) defined. If I didn't miss it, please add a clarification for this definition.

[Author Response · NeurIPS 2019]

We would like to thank the reviewers for their valuable comments. Below, we address the concerns raised.

**Reviewer 1.**
**Q. The assumption in Line 207 seems strong compared to the results in [Shamir, 2018].**
Please note that [S18] considers only one-block ResNets, and as mentioned in Lines 236–239, the condition in Line 207
is always satisfied when $L = 1$. In other words, our paper subsumes and extends the results of [S18].

**Q. The result in Section 3.2 seems like a trivial application of [Shamir, 2018].**
The example in Section 3.2 is not a result of [S18]. Rather, the example is meant to show that direct application of [S18]
is **not possible** for deeper ResNets. As discussed in Lines 178–182, an application of [S18] only proves $\mathrm{err}(H_2, Y) \leq$
$\mathrm{err}(H_1, Y)$, so a comparison of $\mathrm{err}(H_2, Y)$ and $\mathrm{err}(X, Y)$ does not follow from [S18]. Further, our example shows that
even $\mathrm{err}(H_1, Y) > \mathrm{err}(X, Y)$ is possible, showing that theoretically proving $\mathrm{err}(H_2, Y) \leq \mathrm{err}(X, Y)$ is challenging
even for $L = 2$. (We hope that the reviewer's confusion regarding Lines 179–181 is clarified by this answer.)

**Q. Section 3 feels separated from the following results in the paper.**
The examples make the following points: (1) the advantage of ResNets; and (2) the difficulty of analyzing *deep* ResNets.
In particular, the second example highlights that the results in the subsequent sections are not trivial extensions of [S18].

We will also reflect the other comments in the revision.

**Reviewer 2.**
**Q. The paper argues that the geometric conditions [...] sufficient conditions when it should hold.**
Thanks for pointing this out. In fact, we do provide a sufficient architectural condition; please refer to Lines 226–235.

**Reviewer 3.**
**Q. My main concern regarding this paper is that proofs [...] future improvement based on them.**
Our proof is not based on a recursive analysis of each residual block. By recursive analysis, we believe that the reviewer
means some induction-like argument, which is not the case here. We analyze each block of the network separately using
partial derivatives and carefully designed perturbations, and collect the results from each block to prove the theorem.
We should mention that our proof technique already reveals the importance of identity skip connection in ResNets: If
the identity terms in residual blocks are replaced with tunable parameter matrices $\boldsymbol{W}_l$, it is impossible to apply the
argument in Lines 405–409 unless strong assumptions on $\boldsymbol{W}_l$ are made. This suggests that identity skip connections
are indeed the key to the benign loss landscape.

**Q. Moreover, the second assumption made in theorem 2 [...] a rather strong assumption on the architecture.**
We agree that if the condition in Corollary 3 is not satisfied, the second condition may not necessarily hold. We believe
that weakening/removing the condition is a highly nontrivial future challenge. But as noted in Lines 240–246, we'd like
to stress that not only does Theorem 2 subsume previous results on shallow ResNets, but it also shows that a **chain**
**of multiple** skip connections, *as opposed to direct skip connections to the output*, has beneficial effects on the loss
landscape. It has been argued that deep ResNets have benign loss landscapes compared to fully-connected networks,
but a rigorous theoretical understanding is still missing; we believe that our results take a step forward in that direction.

**Q. The examples provided make understanding and reading the paper easier, [...] standalone results.**
We agree; the examples are just to note the advantage of ResNets and highlight the difficulty of analyzing *deep* ResNets.

**Q. Additionally, section 5 strongly relies on the near-identity assumption [...] incremental improvements.**
As summarized in the paper, previous theoretical results have shown that near-identity regions are expressive and have
certain benign optimization properties. In addition, a recent paper *"Fixup Initialization: Residual Learning Without*
*Normalization"* showed that initializing ResNets in near-identity regions also leads to good empirical performance. The
finding of that paper is rather surprising. For each residual part $\Phi_{\boldsymbol{\theta}}^l(\cdot)$ (according to our notation), Fixup initializes the
last layer of $\Phi_{\boldsymbol{\theta}}^l(\cdot)$ at zero, and initializes the other layers by using standard random schemes; and then it multiplies a
factor inversely proportional to depth $L$. This means that each $\Phi_{\boldsymbol{\theta}}^l(\cdot)$ at initialization is zero, hence the network *does*
*start in the near-identity region*. Using this initialization scheme, their experiments demonstrate that ResNets can
be stably trained **without batch normalization**, and trained networks match the **generalization** performance of the
state-of-the-art models. The Fixup paper thus suggests that understanding optimization and generalization of ResNets
in near-identity regions is a meaningful and important problem, thus further motivating our analysis in Section 5.

**Q. Lastly, in line 119 it is claimed that augmenting one dimension [...] Can you please elaborate on this trick?**
We admit that we didn't provide enough details about this trick. The trick requires some additional structures, e.g., the
residual part $\Phi_{\boldsymbol{\theta}}^l$ must have an additional output dimension that always outputs 0. Due to space limits, we refer the
reviewer to Remark 1 of [Shamir, 2018]; we will add more details as we revise our paper.

**Q. Definition of $\mathrm{col}(\cdot)$?**
The notation $\mathrm{col}(A)$ means the column space of a matrix $A$. We will add the definition in the next revision.

[Meta-Review · NeurIPS 2019]

This paper builds upon and extends the work of Shamir'18 in an attempt to study the power of Resnets. They provide several motivating examples and new results regarding deep resnets. While the results here seem to rely on some strong assumptions, overall it seems that the authors are able to make a certain progress using non trivial techniques. Hopefully future work will either show these assumption to be redundant or necessary.